# Economic expenditures by recreational anglers in a recovering atlantic bluefin tuna fishery

Kristian Maar [1]*, Christian Riisager-Simonsen[2], Brian R. MacKenzie[3], Christian Skov[4], Kim Aarestrup[4], Jon C. Svendsen[4,5]

1 Centre for Ocean Life, National Institute of Aquatic Resources, (DTU Aqua), Technical University of Denmark, Kgs. Lyngby, Denmark, 2 Section for Maritime Service, National Institute of Aquatic Resources, (DTU Aqua), Technical University of Denmark, Kgs. Lyngby, Denmark, 3 Section for Oceans and Arctic, National Institute of Aquatic Resources, (DTU Aqua), Technical University of Denmark, Kgs. Lyngby, Denmark, 4 Section for Freshwater Fisheries Ecology, National Institute of Aquatic Resources, (DTU Aqua), Technical University of Denmark, Silkeborg, Denmark, 5 Section for Coastal Ecology, National Institute of Aquatic Resources (DTU Aqua), Technical University of Denmark, Kgs. Lyngby, Denmark

* krmaa@aqua.dtu.dk

**Data Availability Statement:** All relevant data are within the manuscript and its Supporting Information files.

## Abstract

The recent return of Atlantic bluefin tuna to northern Europe following the recovery of the east Atlantic stock has sparked substantial public and scientific interest. This is particularly true for recreational anglers in Denmark, who often consider Atlantic bluefin tuna to be the catch of a lifetime. This attitude has previously sustained a substantial recreational fishery for bluefin tuna with annual tournaments in Denmark, which peaked in the 1950s before the subsequent collapse of the stock during the 1960s. Several scientific tagging programs have recruited recreational anglers in recent years to help catch and release tagged bluefin tuna. The anglers' investment of time and money in the scientific tagging projects indicate that the recreational fishery could recover in the future. However, the economic aspects of a potential future recreational bluefin tuna fishery remain unknown. We surveyed anglers participating in a scientific catch and release bluefin tuna fishery in Denmark across three years (2018–2020) and calculated the total annual expenditures associated with the activities. Additionally, we estimated the magnitude of the negative impact (i.e., incidental mortalities) on the bluefin tuna stock. Our results show that total annual expenditures by the recreational anglers approached 1,439,540€, totaling 4,318,620€ between 2018 and 2020. We found that recreational bluefin tuna anglers had mean annual expenditures directly related to the bluefin tuna fishing between 7,047€ and 2,176€ with an associated mortality impact on the stock of less than 1 tonne annually. By comparing the mortality impact to the expenditures, we estimate that each dead Atlantic bluefin tuna during the three study years generated 398,163€ in mean annual expenditures, equivalent to approximately 1636€ kg$^{-1}$. Our study demonstrates significant economic expenditures among recreational anglers who target Atlantic bluefin tuna. This provides a clear example of how a recovery of marine natural capital and related ecosystem services can support development in the blue economy.

**Funding:** Funding was provided by the EU interreg project MarGen II, Danish rod and net license funds, the Nordic Council of Ministers (project nos. 141-2016-Tuna; 180-2018-Tuna tagging), European Maritime and Fisheries Fund, and the Hempel Foundation. The funders had no role in study design, data collection and analysis, decision to publish, or preparation of the manuscript.

# Introduction

The Atlantic bluefin tuna (*Thunnus thynnus*; ABFT) is an iconic and highly migratory species, which is among the most valuable fish in both commercial and recreational fisheries [1]. ABFT supported seasonal commercial and recreational fisheries near Denmark and in other parts of Scandinavia in the 1940s and 1950s, but by the first half of the 1960s, these fisheries collapsed, [2, 3]. Furthermore, the increasing global market demand for ABFT and resulting exploitation during the 1990s to early 2000s reduced ABFT biomass in the entire stock area (northeast Atlantic and Mediterranean) and raised concerns that the stock could collapse if exploitation was not reduced [4, 5]. The ABFT fishery is currently managed through the International Commission for the Conservation of Atlantic Tunas (ICCAT). To allow the stock to recover, ICCAT introduced a 15-year recovery plan in 2007 with reduced quotas for Contracting Parties [6]. Subsequent stock assessments have indicated a recovery of the stock with associated increases in quotas for the ICCAT member states [3]. In 2018, ICCAT adopted a management plan replacing the recovery plan [7], signifying a change in management strategy for the stock from recovery to sustainable exploitation. The recovery of ABFT has also been officially recognized by The International Union for Conservation of Nature (IUCN), which changed the status of ABFT from "Endangered" to "Least Concern" in September 2021 [8]. Following the recovery, ABFT has become more common in the northern part of its range, and, in 2016 and all subsequent years, several sightings of ABFT have been reported in Denmark [9] following more than 50 years when the species was seldom observed. In spite of considerable commercial and recreational fisheries for ABFT in Denmark up until 1950s [10], the fisheries had collapsed by the time the first quotas were allocated through ICCAT in 1992. Denmark and some neighboring countries (e. g., Sweden) therefore presently lack an annual quota allocation for either commercial or recreational fishing.

Nevertheless, the return of ABFT to Danish waters has sparked significant scientific and public interest. Similarly, the commercial fishing sector in Denmark has indicated an interest [11] in developing a consumption ABFT fishery if an adequate quota can be obtained in the future.

The importance of ABFT for the commercial fisheries sector is well established [e.g. 1], as also evidenced by global export-import arrangements for trade in the species. However, from an economic perspective there are indications that recreational fishing may provide larger economic benefits if ABFT are exploited by recreational anglers as a cultural ecosystem service rather than as for food (i.e. a provisioning ecosystem service) [12]. In Canada for example, recreational charter industry based on catch and release angling for ABFT was estimated to generate six times more revenue per tonne (t) compared to harvest based commercial fishing [13]. The global recovery of the stock has also coincided with a decrease in the commercial price at the final point of sale of ABFT, which has decreased by 47% from $67 kg$^{-1}$ in 2012 to $37 kg$^{-1}$ in 2018 [1]. While the recovery has resulted in increased quotas and tripled the total reported landings from 13,000 t in 2012 to 30,000 t in 2018, the end value of the commercial ABFT fishing sector has only grown from $870 million to $1.1 billion [1].

## The scientific tag and release program (T&R program)

To understand the biological background for the recovery of ABFT in Scandinavia, a research project was carried out with tagging of tuna in the Skagerrak between 2017 and 2021. The project adopted a method for tagging ABFT involving rod and reel capture, similar to methods used by catch and release fisheries in the United States of America and Canada [14]. Voluntary anglers with adequate boats, gear and experience were recruited for the T&R program to catch ABFT, using conventional rod and reel tackle. More anglers than needed for the T&R program

**Table 1. Mortalities of ABFT caught and released using rod and reel.** [14, 16–22]. Mortality in % is calculated based on the number of ABFT tagged minus the number of nonreports divided by the number of mortalities. Mortality categories include mortalities immediate (e.g. at-vessel), post-release mortality, both or none if no mortalities were observed. Both immediate and post-release mortality were counted when calculating the mortality percentages.

| | ABFT tagged (#) | ABFT without non-report | Mortalities (#) | Mortality (%) | Mortality category | Non-report (fate unknown) |
|---|---|---|---|---|---|---|
| Block et al. (1998) | 37 | 35 | 0 | 0 | None | 2 |
| Lutcavage et al. (1999) | 20 | 17 | 0 | 0 | None | 3 |
| Stokesbury et al. (2004) | 35 | 32 | 1 | 3.1 | Immediate | 3 |
| Wilson et al. (2005) | 68 | 60 | 1 | 1.7 | Immediate | 8 |
| Stokesbury et al. (2007) | 6 | 3 | 0 | 0 | None | 3 |
| Galuardi et al. (2010) | 41 | 36 | 0 | 0 | None | 5 |
| Stokesbury et al. (2011) | 60 | 56 | 3 | 5.4 | Both | 4 |
| Marcek & Graves (2014) | 20 | 20 | 0 | 0 | Post release | 0 |

volunteered highlighting the high motivation in the angling community to fish for ABFT. The aim of the scientific T&R program was to gather data on the biology and migratory behavior of ABFT passing through Scandinavian waters.

From the anglers' perspective, the fishing activities within the T&R program mimic catch and release fishing for ABFT in e.g. Croatia, the U.S.A, or Canada. This non-consumptive orientation is characteristic of ABFT anglers, and the more central ABFT fishing is to their lifestyle, the more likely they are to have a positive attitude towards catch and release fishing for ABFT [15]. Catch and release fishing for ABFT relies on the experience of catching ABFT as the primary source of interest and associated monetary spending, and mortality of ABFT is incidental rather than intentional. Incidental mortality of catch and release ABFT angling using rod and reel has been found to be low, typically ranging from 0% to 5% (Table 1), enabling recreational catch and release to have a minimal impact on the ABFT population.

Similar to the assessment of many other marine cultural ecosystem services [23], the economic impacts related to ABFT angling in Scandinavia is presently unknown, but could be important for the region's economy, comparable to the situations in North America [24, 25]. To estimate the economic potential of a future recreational ABFT fishery in Scandinavia, we therefore studied angler behavior and expenditures associated with the recreational fishing involved in the Scandinavian T&R program across three years (2018–2020). Specifically, the aim of this study was to determine the economic potential in the recovering recreational ABFT fishery in Scandinavia by evaluating the expenditures of recreational ABFT anglers associated with the T&R program. To this end, the Scandinavian T&R program was studied as a proxy for a future catch and release program.

## Methods

### "Ethics statement

The study involved interviews with volunteers and gathered information was kept confidentially. No medical records or archived samples were used in our study. All data was anonymized. Our study included no patients. No animals were used in our study as our study was concerned with economic expenditures of people involved in recreational activities. No anesthesia, euthanasia or any kind of animal sacrifice was covered by our study. Given this set-up for the research, we decided that committee approval for our study was not necessary."

Recreational *angling* is in this context defined as a form of recreational fishing in which participants, known as anglers, use fishing rod, reels, hook and line to catch fish. *Catch and release* also known as *live release* angling is a mode of recreational angling where some or all of the

fish caught are released alive and the aim of the activity is partly or exclusively experiential rather than consumptive. Motivations for recreational angling includes challenge, achievement, sport, recreation, relaxation, social activity etc. [26].

**Angler participation in the T&R program.** The Scandinavian fishing for ABFT took place exclusively as part of a T&R program which aimed to capture, tag and release ABFT [27]. The fishery was organized by a research team from the Technical University of Denmark who directed the fishing activities, including the timing and duration of each season (defined as the time from August-September in each year in which ABFT could take place), the fishing area, minimum gear and vessel requirements and number of participants. The ABFT fishery was therefore confined to a specific group of official participants, with no other ABFT fishing allowed in Denmark. Formal permission for the fishery was obtained from ICCAT and relevant national authorities in Denmark; the latter were also responsible for ensuring compliance with the fishing regulations. The framework adopted to manage the fishery in the context of the T&R program was largely based on the recreational fishing regulations for catch and release fishing for ABFT in Canada [28] and included the exclusive use of circle hooks, and appropriately sized fishing gear. The fishing tackle used for ABFT fishing, including rods, reels, line, hooks etc. are highly specialized. Anglers were recruited for the T&R program based on their previous experience fishing for ABFT or other large game fish species (e. g., yellowfin tuna (*Thunnus albacares*), and billfish (*Istiophoridae*)) in other parts of the world. Only anglers adhering to minimum gear requirements (e.g. the use of circle hooks and 130 lbs. test line), and with appropriate vessels for the type of fishing were allowed to participate. Participating anglers were responsible for their own expenses related to their participation in the ABFT fishery. Anglers decided themselves how much time and money they wished to allocate to fishing for ABFT as long as the minimum gear requirements were met. There are no other recreational fisheries in Denmark which rely on the specialized equipment and fishing methods used to fish for ABFT. For this reason, anglers cannot reuse equipment they already own and use for e.g. salmon fishing. The anglers would thus have to invest in specialized tuna fishing gear to participate in the fishery even if no minimum gear requirements were enforced by the tagging project. Therefore, we assume that the expenditures related to ABFT fishing within the T&R program reflect expenditures that would have occurred if there had been an open recreational ABFT catch and release fishery of comparable size and format as those in other regions (e.g. Canada, U.S.A or Croatia).

The COVID-19 pandemic was ongoing during the ABFT fishing activities under the T&R program in Denmark in 2020. The fishing took place in August-September at a time with only few restrictions on the free movement and behavior of residents in Denmark, and no restrictions related to the pandemic applied specifically to the ABFT anglers or the fishery.

**Survey.** The expenditure survey (See S1 File) was designed based on the Recreational Fisheries Economic Impact Assessment Manual published by the Food and Agriculture Organization of the United Nations [29]. We developed the methodology and survey based on previous expenditure surveys applied to recreational fisheries [30–33]. Questions addressed individual recreational anglers' monetary expenditures specifically related to the ABFT angling including travel, accommodation and investments in angling equipment. Additional questions addressed personal characteristics and behavior of the respondents, including age, time spent angling for ABFT, travel, and gross annual income. The survey targeted all ABFT anglers with permits to fish for ABFT from 2018–2020, which amounted to 302, 500 and 600 ABFT anglers (sampling units) in 2018, 2019 and 2020 respectively. Previous studies [30, 31, 34, 35] have highlighted that this type of survey represents an effective method of obtaining economic data in similar contexts.

To administer the survey via email to ABFT anglers in Denmark, it was necessary to establish contact with them individually. To obtain their email addresses we attempted to contact anglers in three different ways. 1) Face-to-face at the point of access during the fishing (harbor), 2) via telephone numbers provided by the T&R program, and 3) via an open call in a Facebook group for Scandinavian ABFT anglers. We were only able to perform face-to-face contact at the point of access in 2018 due to logistical limitations in 2019 and 2020. Upon contact, ABFT anglers were asked if they were willing to participate in the survey and, if they were, we asked an email address to which we subsequently sent the survey. There was full and unconditional consent from all interviewees and all contributions were delivered voluntarily. There was no reward for the participation in our study. Information was gathered both verbally and in written form. No minors were included in any parts of the study, including the interviews. In 2018, 2019 and 2020 the questionnaire was sent out immediately after the angling ended (September-October) to reduce recall bias [29]. Two weeks after administering the surveys, a SMS reminder to fill out the survey was sent to the participants. Partial survey responses were discarded. Respondents reported expenditures in Danish kr. (DKR) and an exchange rate of 7.45 DKR €$^{-1}$ was used to convert the expenditures to euros (€).

**Angler categories.** Our approach corresponds to the approach used by Morales-Nin et al. [33], and we separated respondents into two groups, as anglers owning or co-owning boats and anglers having no ownership in the boats (hereafter "regular" anglers) have significantly different costs related to angling. The total number of anglers owning boats and regular anglers in each year was provided through the registration to the T&R project, and therefore the relative proportion of regular anglers to anglers owning boats could be determined for each year.

**Behavior.** Mean values of time spent ABFT fishing, mean distance travelled related to ABFT fishing, annual income and similar metrics available via the survey were used to reveal demographic and behavioral characteristics.

**Expenditures.** The expenditures reported by the survey respondents were raised to the total number of permit holders in the recreational fishery for ABFT by multiplying the annual mean expenditures of regular anglers and anglers owning boats separately with the total number of regular anglers and anglers owning boats participating in the recreational tuna fishery in each year (2018–2020).

**Impact on the stock.** The ICCAT management plan [3] for ABFT regulates the total allowable catch in tonnes and includes estimates of the entire biomass in the stock area (northeast Atlantic Ocean and Mediterranean Sea). The biomass of ABFT due to incidental mortality from a recreational catch and release fishery will give an indication of its impact on the stock. Researchers from the T&R program recorded immediate (observed) mortalities during the fishing activities as well as known and presumed post-release mortalities (based on tagging data and individual condition upon release). We calculated a relative mortality rate of recreationally caught ABFT by dividing the total number of immediate and post-release ABFT mortalities in the T&R fishery by the total number of ABFT caught each year. We derived upper and lower confidence limits (Cl) of the mortality estimates in each year using the Clopper-Pearson binomial proportion method [37]. The resulting mortality of ABFT in the recreational fishery applied in our calculations was 4.35% (95% CL: 2.54, 6.97). We multiplied the mortality estimates with the total number of ABFT caught in each year to estimate the number of ABFT that may have died incidentally in each year as a direct result of the recreational angling and scientific tagging activity. The estimated absolute number of ABFT lost due to incidental mortality was multiplied with the mean weight of ABFT caught in each respective year to estimate the mortality impact on stock biomass. All tunas were length measured on a platform on the stern of a tagging vessel by researchers from the T&R program: lengths of tunas caught were measured as curved fork length (CFL). We converted measures of individual curved fork

length (to nearest 1 cm) to straight fork length (SFL) using SFL = 1.7959 + 0.9517 x CFL and estimated individual body mass in kg (W) using W = 0.0000350801 x $SFL^{2.878501}$ [38]. The status and condition of the tunas during measuring and tagging were closely monitored by the tagging crew and health indicators including bleeding, inactivity or injury were recorded. These observations were subsequently used to evaluate the likelihood of post release mortality caused by the catching and subsequent release of the tunas. Tunas with significant bleeding, injury or inactivity upon release were presumed dead. The results were used to estimate the total mass of the tunas that died incidentally as a consequence of the recreational angling on an annual basis. Expenditures $tuna^{-1}$ and $kg^{-1}$ of tuna due to mortality were calculated by dividing total expenditures in each year with the estimated total number and total biomass of ABFT, respectively, due to incidental mortality within the same year. The annual values of expenditures $tuna^{-1}$ and $kg^{-1}$ were then averaged across years to derive an estimate for the 3-year period. In addition, we summarized the literature for reported mortalities of ABFT caught using recreational fishing methods (Table 1).

## Results

### Expenditures

A total of 302, 500 and 600 ABFT anglers were registered as participants in the fishery in 2018, 2019 and 2020 respectively. Of these a total of 219 agreed to participate in the expenditure survey. The survey yielded a total of 131 valid survey responses with a response rate of 57%, 47% and 72% in 2018, 2019 and 2020, respectively (Table 2). The total estimated expenditures of ABFT anglers in Denmark from 2018–2020 was 4,318,620€, resulting in total average annual expenditures of 1,439,540€. Individual anglers owning boats spent an annual average of 7,047 € on ABFT fishing, which is substantially more than individual regular anglers not owning

**Table 2. Key results of the from the survey of ABFT anglers from 2018–2020 including total numbers of participants in the ABFT fishery in the categories of regular anglers and anglers owning boats.** Mean and total annual expenditures in Euro for regular anglers and anglers owning boats with lower and upper [L U] 95% Confidence limits (Cl) for expenditures based on the mortality estimates derived according to Clopper and Pearson (1934) [36]. Mortality includes both immediate and post-release mortality.

| | 2018 | Cl 95% [L U] | 2019 | Cl 95% [L U] | 2020 | Cl 95% [L U] | | Cl 95% [L U] |
|---|---|---|---|---|---|---|---|---|
| Total number of participants | 302 | - | 500 | - | 600 | - | | - |
| Regular anglers | 227 | - | 400 | - | 475 | - | | - |
| Anglers owning boats | 75 | - | 100 | - | 125 | - | **Total** | - |
| Participants interviewed | 53 | - | 83 | - | 83 | - | 219 | - |
| Survey responses regular anglers | 12 | - | 11 | - | 21 | - | 44 | - |
| Survey responses anglers owning boats | 18 | - | 30 | - | 39 | - | 87 | - |
| Survey responses (total) | 30 | - | 41 | - | 60 | - | 131 | - |
| Response rate | 57% | - | 49% | - | 72% | - | | |
| | | | | | | | **Mean** | |
| Number of tuna caught | 104 | - | 56 | - | 116 | - | 92 | - |
| Mean individual weight of tunas caught (kg) | 234 | - | 240 | - | 260 | - | 245 | - |
| Mortality total (%) | 2.88 | [0.75, 6.96] | 5.36 | [1.66, 14.78] | 5.17 | [2.28, 9.95] | 4.35 | [2.54, 6.97] |
| Mortalities (# of deceased tuna) | 4.5 | [2.64, 7.25] | 2.4 | [1.42, 3.90] | 5.1 | [2.95, 9.09] | 4 | [2.34, 6.41] |
| Incidental mortality (t) | 1.1 | [0.6, 1.7] | 0.6 | [0.3, 1.0] | 1.3 | 0.8, 2.1 | 1.0 | [0.6, 1.6] |
| Mean expenditures (€) regular anglers | 2,356 | - | 2,355 | - | 1,817 | - | 2,176 | - |
| Mean expenditures (€) anglers owning boats | 10,734 | - | 5,085 | - | 5,321 | - | 7,047 | - |
| Expenditures (€) $kg^{-1}$ | 1,263 | [788, 2,163,] | 2,481 | [1,549, 4,249] | 1,080 | [727, 1,995] | 1,636 | [949, 2,516] |
| Expenditures (€) dead $tuna^{-1}$ | 296,173 | [184,843, 507,226] | 595,465 | [371,631, 1,19,792] | 302,850 | [189,10, 518,661] | 398,163 | [224,493, 616,31] |

**Table 3. Total annual expenditures in EUR by category for regular anglers and anglers owning boats exclusively related to recreational fishing for ABFT as part of the T&R program.**

| Category | 2018 | | 2019 | | 2020 | |
|---|---|---|---|---|---|---|
| | Anglers owning boats | Regular anglers | Anglers owning boats | Regular anglers | Anglers owning boats | Regular anglers |
| Angling gear | 192,796 | 125,757 | 141,119 | 365,324 | 176,774 | 155,026 |
| Tackle | 66,421 | 133,557 | 52,421 | 70,861 | 79,533 | 86,934 |
| Bait | 6,946 | 19,501 | 1,790 | 6,935 | 1,874 | 7,661 |
| Boat expenses | 212,763 | 6,925 | 60,913 | 91,107 | 106,199 | 91,539 |
| Boat equipment | 166,672 | 1,108 | 89,485 | 65,884 | 59,033 | 58,546 |
| Harbor costs | 8,781 | 19,667 | 17,173 | 24,279 | 34,291 | 25,154 |
| Other equipment | 6,890 | 16,897 | 9,253 | 22,707 | 17,192 | 50,976 |
| Fuel | 77,297 | 95,538 | 53,505 | 120,176 | 75,459 | 111,929 |
| Accommodation | 31,482 | 27,700 | 25,378 | 36,353 | 35,470 | 75,609 |
| Transport | 12,930 | 20,775 | 11,329 | 32,550 | 27,121 | 33,907 |
| Public transport | 1,398 | 2,770 | 0 | 5,034 | 172 | 6,376 |
| Souvenirs | 979 | 3,739 | 3,284 | 8,166 | 3,599 | 3,289 |
| Food and drink | 15,296 | 53,184 | 25,982 | 80,397 | 31,109 | 116,556 |
| Additional | 4,362 | 7,756 | 16,911 | 12,237 | 17,318 | 39,533 |
| Total expenditures | 1,339,887 | | 1,450,552 | | 1,528,182 | |

boats, who spent an average of 2,176€ annually. Due to the higher total number of regular anglers, the total annual spending was higher for regular anglers at 779,974€ than anglers owning boats who had total annual spending of 659,567€. The largest categories of spending was fishing gear, boat expenses and fuel accounting for 27%, 13% and 12% of total expenditures, respectively (Table 3).

## Demographics

The mean age of all respondents was 51, 51 and 52 years of age in 2018, 2019 and 2020, respectively. The anglers were predominantly male with 97%, 93% and 93% of respondents identifying as male in 2018, 2019 and 2020, respectively. Anglers owning boats and regular anglers had mean annual income before taxes of 85,748€ and 84,064€; consequently, anglers owning boats spent on average 8.2% of their annual pre-tax income on recreational ABFT fishing while regular anglers spent 2.6%. The mean value of the individual boats participating in the fishery was 73,467€. Anglers with and without boats were predominantly (92%) non-local residents, here defined as those living outside the postal zip code of the point of access (Skagen) or a neighboring zip code to the point of access for the fishing. This pattern is also reflected in the mean distance travelled by each angler related to the fishery reaching 1,213 km annually.

## Angler behavior

In 2018 and 2019, the recreational angling year for ABFT was limited to a 14-day period, whereas in 2020, the angling year was limited to a 16-day period. Weather conditions restricted the fishing within these periods with anglers being prevented from fishing due to weather an average of three days in 2018, and seven days in both 2019 and 2020. Anglers reported fishing for ABFT an average of six days in 2018 and 2019 and an average of five days in 2020, spending an average 9.9 hours fishing pr. fishing day. Anglers reported sailing an average of 340 nautical miles pr. year during the ABFT fishing. Anglers predominantly chose to stay in the Skagen area on bad weather days, spending an average of nine days in Skagen (entry point to the fishery) pr. season in all three years. In addition, anglers spent 70, 75 and 52

hours on preparation before the fishing activities in 2018, 2019 and 2020, respectively. No respondents reported any time spent targeting other species of fish during the period apart from catching baitfish intended for ABFT fishing. The average number of anglers on board each boat was three in 2018 and four in 2019 and 2020. In total, 73% of anglers reported experiencing catching ABFT personally, or as part of a team effort, in 2018 and 2019 whereas 77% did so in 2020. The corresponding number of ABFT caught pr. angler pr. season was 1.9, 1.5 and 2.1 in 2018, 2019 and 2020 respectively. However, multiple anglers usually take part in catching each ABFT, and so more than one angler experienced catching the same ABFT (although the ABFT was only caught once).

## Impact on the stock

A total of 12 ABFT were lost due to incidental mortality caused by the recreational fishery. The total number of ABFT caught from 2018–2020 was 276 fish, giving a mortality of 4.35% (95% CI = 2.54, 6.97%, corresponding to a range of 7–19 individuals). Of the 276 ABFT caught, 104, 56 and 116 were caught in 2018, 2019 and 2020, respectively, giving an average annual catch of 92 ABFT. The mean size of ABFT caught in each year was 245 cm, 247 cm and 254 cm (CFL) corresponding to weights of 234 kg, 240 kg and 260 kg in 2018, 2019 and 2020, respectively. The resulting biomass of ABFT attributed to incidental mortality related to the recreational fishery and associated tagging efforts from 2018–2020 is 1 tonne annually (95% CL = 0.6, 1.6).

## Discussion

We surveyed recreational anglers participating in a scientific ABFT fishery and analyzed their expenditures in the period from 2018–2020. Our survey design was based on previous studies of economic expenditures in recreational fishing and we applied it to a substantial recreational ABFT fishing sector which has supported the T&R program in Denmark. The first ABFT was caught in Denmark in 2017 after nearly 60 years of absence, and our results indicate that the angling sector has responded with a dramatic increase in number of ABFT anglers and spending associated to the fishery during the subsequent years. Over a period of three years we contacted 219 ABFT anglers and received 131 expenditure survey responses from both regular anglers and anglers who owned or part-owned the boats from which the fishing was performed.

Our survey revealed an annual average of 1,439,540€ of expenditures associated with ABFT angling during the period. In comparison the entire Danish angling sector without ABFT fishing totals 387€ million annually but the average angler only spends 336€ annually [39] which is six times less than the amount spent on ABFT angling by regular anglers and 20 times less by anglers owning boats. This indicates that fishing recreationally for ABFT has much higher associated costs than the average type of recreational fishing activities in Denmark.

Our study indicates that Danish ABFT anglers spent an average of 2.5–8.2% of their annual pre-tax income on recreational ABFT fishing and allocated an average of 10 days per year in order to participate in the fishery. Both numbers suggest strong motivation among participating anglers for ABFT fishing. Expenditures in ABFT fishing are even higher than boat fishing for Atlantic salmon (*Salmo salar*) and sea run brown trout (*Salmo trutta*) which has traditionally had the highest levels of expenditures by anglers in Denmark [39]. Anglers participating in salmon and sea run sea trout fishing had annual expenditures of 2826€ angler[-1] equivalent to 6.2% of their annual income [39] which is slightly more than the 2,176€ spent annually on ABFT by regular anglers, but less than half of the amount spent on ABFT by anglers owning boats. This makes ABFT the type of fishing in Denmark with the highest associated annual expenditures by anglers.

When sending out the questionnaire we had no knowledge whether the anglers had ownership in a boat or were regular anglers. Unfortunately, we received a lower number of responses from regular anglers than anglers owning boats and the sample size for the regular anglers is lower than we hoped, totaling 44 responses and with 2018 having the lowest number of 11 responses. We attempted to recruit more regular anglers to respond to the survey through the open Facebook call but received only four responses. One explanation for the bias in number of responses towards boat owners is that they are more interested in ABFT fishing in general and, as a consequence, were more likely to respond to the survey. As we separate the groups of regular anglers and anglers owning boats in our analysis, we therefore see higher confidence in the expenditures for boat owners compared to regular anglers. As the sample size for regular anglers is small, there is a possibility that the actual total expenditures in the group of regular anglers could be lower or higher than those we found.

Rather than relying on ABFT as a consumable commodity, a recreational catch and release fishery is based on the value anglers are willing to attribute to the activity. This attributable value is experiential rather than consumptive, and likely has both an individual as well as a social character for the involved recreational fishers, given that multiple anglers typically participated on the same boat in catching the same individual ABFT during the effort to get the tuna to the boat. Similarly this indicates that most or all the expenditures paid by multiple anglers is derived from the experience of catching relatively few ABFT. As a result, it is relevant to scale the economic impact of the fishery with the number of participants acquiring an experience of fishing for and catching an ABFT rather than the number of tunas caught. In the commercial fishery for Eastern Atlantic BFT higher landings does not necessarily mean higher revenue as a rapid increase in total allowable catch resulting in an increase in supply will negatively affect the price of ABFT as a market commodity [40].

The mortality resulting from recreational catch and release fishing for ABFT was 4.35%, which is comparable to mortalities reported in the literature for this type of fishing activity. By comparing the mortality and the expenditures, each deceased ABFT was on average associated with 398,163€ in expenditures equivalent to 1,636€ kg$^{-1}$1. Expenditures kg$^{-1}$ of deceased ABFT is four times higher than trout and salmon which are the most popular target species of recreational angling in Denmark constituting 47% of the Danish angling sector [39], which generate expenditures of 353€ kg$^{-1}$ [41].

The recovery of ABFT in Scandinavian waters represents a similar economic opportunity as the recreational ABFT fishery which developed near Hatteras, North Carolina in the 1990s. In 1996, this newly emerged fishery had 2,900 boat trips targeting ABFT where there had been none just a few years before [25]. In 1997, just three years after the fishery emerged near Hatteras, expenditures by ABFT anglers resulted in a total impact of $4,627,108 and $5,032,870 on the Hatteras-area economy and North Carolina economy, respectively, resulting in 126 full and part-time jobs attributed to this fishery in 1997 [25]. The expenditures we present here are directly related to the activity of fishing recreationally for ABFT. As this activity has not previously been possible in Denmark for several decades due to the long period of rarity of ABFT and lack of quota, it follows that the related expenditures represent mainly new economic activity.

Anglers were not limited by restrictions due to COVID-19 during their participation in ABFT fishing in the 2020 season and it is likely that their related expenditures where not inhibited during the period. On the contrary, the lack of available options for recreation in the time up to the ABFT season e.g. travelling abroad for holidays, may have restricted spending on other goods and services, allowing savings to be spent on ABFT fishing instead, resulting in higher spending than under normal conditions.

We assumed an averaged overall mortality of 4.35% for the captured and released tunas to derive an estimate of the expenditure per deceased tuna, and to enable comparison of this indicator with those in other regions where ABFT is exploited in catch and release or tag and release fisheries. The fish caught in the Danish tagging project were captured, tagged, sampled and released. It is possible that mortalities in such a fishery could be higher than in a catch-release fishery without tagging. However, most rod-reel based estimates of ABFT mortality have been obtained in catch-tag-release fisheries. In such fisheries, ABFT are often handled more after capture e.g. mounting tags, taking tissue and blood samples etc. [14, 21, 42] than would be common in a catch and release recreational fishery without tagging. ABFT mortalities may depend on the angling method. In particular, the use of circle hooks lowers the mortality [43]. If the anglers follow proper handling practices, it is therefore likely that incidental mortality in a catch and release recreational fishery could be lower than the numbers reported in scientific tagging studies (Table 1).

Assuming a 4.35% mortality, the total estimated loss of ABFT due to incidental mortality during the study period of three years was 3t, (i.e. one t per year 95% CL = 0.6, 1.6). When comparing the annual mortalities here, including the upper 95% CI estimate, with the total allowable catch of ABFT in the northeast Atlantic of 36,000t in 2020–2022 [10], it is evident that a substantial recreational sector could develop in Denmark and likely other parts of Scandinavia if a small quota was allocated for this purpose.

In support of previous studies [25, 44], the study indicates that recreational angling for ABFT presents an emerging opportunity for economic development in the areas where tuna populations have recovered. We expect that the future potential for economic activity in a recreational ABFT fishery will likely be much higher if the sector can develop beyond the scope of scientific tagging projects. There are multiple restrictions enforced by the tagging projects which likely decrease the level of expenditures and participation associated with recreational ABFT fishing. 1) The number of anglers is limited by the tagging project operations, and in all years, several anglers were not able to participate even though they indicated their interest by applying for the tagging projects. 2) The fishing season is restricted by the tagging projects, and in this case, anglers were only allowed to fish for ABFT a maximum of 14 days in 2018–2019 and 21 days in 2020. Even though anglers may prioritize their participation in the fishery within this period, many anglers indicated that they would fish for ABFT beyond the timeframe of the tagging projects if allowed to do so. 3) The tagging projects restricted the possibility for charter operations to offer ABFT fishing as a service. Charter fishing for ABFT could likely develop into a significant sector if allowed to do so. 4) The tagging projects recruit almost exclusively local anglers. This, in addition to the lack of a charter fleet, effectively inhibits the development of a tourism sector based on recreational ABFT fishing. Big game fishing including fishing for ABFT is an important sector of tourism demonstrated by the fishery in North Carolina (U.S.A.) where just 15.6% of anglers where local residents, and charter trips outnumbered private trips by nearly a factor of three [25]. A selection criteria for participating in the ABFT fishery was experience in ABFT or similar big game fishing which means that the Danish anglers themselves have previously traveled abroad for this activity. In addition to increasing economic activity by attracting non-local anglers and thus increasing the total number of participants in the fishery, non-locals also generate more new revenue in the local economy as opposed to locals who to a greater degree could be expected to shift their expenditures from one sector (or fish species) to another. From a policy perspective, the present study provides a successful example of how marine conservation actions aimed at restoring lost or reduced biodiversity (e. g., the ICCAT recovery plan), could support economic development, which is a combined policy target in several European strategies related to the Blue Economy and biodiversity protection [45, 46]. Additionally the economic estimates suggests that utilization of

ABFT as provider of primarily cultural ecosystem services (recreational experiences) rather than provisioning service (commercial harvest of food), may be preferable from an economic development perspective, which resonates with policy goals related to decoupling economic growth from biodiversity exploitation and loss, as highlighted by e.g. the European Green Deal [46]. The substantial spending by ABFT anglers across a variety of goods and services combined with low mortality associated with recreational catch and release methods allows economic activity with a low impact on the ABFT stock in the form of incidental mortality. The present study showcases the potential economic benefits associated with recovering iconic marine fish stocks by providing the basis for low impact recreational angling.

## Conclusions

This study indicates that recreational catch and release angling for ABFT in Northern Europe presents an attractive economic opportunity with low associated mortality and impact on the stock. The high level of effort and expenditures required to participate in the fishery under the management of the T&R program demonstrate a basis for the potential development of a considerable recreational ABFT fishing sector, assuming quotas are available. These data provide a new scientific and economic basis for policy makers and managers seeking to manage the different options for exploiting ABFT fisheries in Northern Europe sustainably. Presently the case of ABFT recovery provide an example of likely economic benefits from successful restoration of marine biodiversity and its related ecosystem services.

## Supporting information

**S1 File.**
(DOCX)

**S2 File.**
(XLSX)

## Acknowledgments

The authors would like to thank the anglers who participated in the survey and helped in the facilitation and logistics of the study, Prof. Anders Nielsen, DTU Aqua, for assistance with statistical analyses, and the reviewers for constructive comments on the manuscript.

## Author Contributions

**Conceptualization:** Kristian Maar, Christian Riisager-Simonsen, Brian R. MacKenzie, Christian Skov, Kim Aarestrup, Jon C. Svendsen.

**Data curation:** Kristian Maar.

**Formal analysis:** Kristian Maar, Brian R. MacKenzie, Christian Skov, Kim Aarestrup.

**Funding acquisition:** Brian R. MacKenzie, Kim Aarestrup, Jon C. Svendsen.

**Investigation:** Kristian Maar.

**Methodology:** Kristian Maar, Christian Riisager-Simonsen, Christian Skov, Kim Aarestrup, Jon C. Svendsen.

**Project administration:** Kristian Maar, Brian R. MacKenzie, Christian Skov, Kim Aarestrup, Jon C. Svendsen.

**Resources:** Kristian Maar, Jon C. Svendsen.

**Supervision:** Brian R. MacKenzie, Christian Skov, Kim Aarestrup, Jon C. Svendsen.

**Validation:** Christian Riisager-Simonsen, Brian R. MacKenzie.

**Visualization:** Kristian Maar.

**Writing – original draft:** Kristian Maar, Christian Riisager-Simonsen.

**Writing – review & editing:** Christian Riisager-Simonsen, Brian R. MacKenzie, Christian Skov, Kim Aarestrup, Jon C. Svendsen.

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
