## [Decision Letter · Decision Letter 0]

3 Feb 2022

PONE-D-21-36689Economic expenditures by recreational anglers in a recovering Atlantic bluefin tuna fisheryPLOS ONE

Dear Dr. Maar,

Thank you for submitting your manuscript to PLOS ONE. After careful consideration, we feel that it has merit but does not fully meet PLOS ONE’s publication criteria as it currently stands. Therefore, we invite you to submit a revised version of the manuscript that addresses the points raised during the review process.

From the economics perspective, there is confusion between economic value and economic expenditure. Other technical problems concerning fisheries economics have been reported by one reviewer.

Another reviewer has raised several issues about the estimation of mortality rates and its use, and recommends clarification about mortality (e.g., types and sources of mortality). The paper could be improved by including a statistical framework.

The manuscript needs a number of additional minor revisions and a careful revision of the English language.

We look forward to receiving your revised manuscript.

Kind regards,

Antonio Medina Guerrero, Ph.D.

Academic Editor

PLOS ONE

Journal Requirements:

3. We note that Figure 1 in your submission contain map images which may be copyrighted. All PLOS content is published under the Creative Commons Attribution License (CC BY 4.0), which means that the manuscript, images, and Supporting Information files will be freely available online, and any third party is permitted to access, download, copy, distribute, and use these materials in any way, even commercially, with proper attribution. For these reasons, we cannot publish previously copyrighted maps or satellite images created using proprietary data, such as Google software (Google Maps, Street View, and Earth). For more information, see our copyright guidelines: http://journals.plos.org/plosone/s/licenses-and-copyright.

Reviewers' comments:

Reviewer's Responses to Questions

**Comments to the Author**

1. Is the manuscript technically sound, and do the data support the conclusions?

Reviewer #1: Yes

Reviewer #2: Partly

2. Has the statistical analysis been performed appropriately and rigorously? 

Reviewer #1: No

Reviewer #2: Yes

3. Have the authors made all data underlying the findings in their manuscript fully available?

Reviewer #1: Yes

Reviewer #2: Yes

4. Is the manuscript presented in an intelligible fashion and written in standard English?

Reviewer #1: Yes

Reviewer #2: Yes

5. Review Comments to the Author

Reviewer #1: General comments

The document presents an estimation of the potential expenditures of a recreational catch and release fishery for Atlantic bluefin tuna and the amount of bluefin tuna mortality it would imply.

The procedure is simple (an estimation of the average expense multiplied by the number of anglers and relate it to the number and weight of tuna estimated to die in the catch, tagging and release operations) and the conclusions are clear. I am not familiar with this type of studies but, in spite of its simplicity, it provides a clear an interesting message: catch and release fisheries can be a source of income more profitable than extraction fisheries by providing an alternative type (non-lethal) of ecosystem services.

In general, there are only a couple of moderately important issues that need further explanation and exploration:

- Mortality: There are usually two types of mortality in tag and release studies: an immediate mortality (the fish is dead, or almost dead when hauled on-board) and a post-release mortality (that takes place when a fish is released seemingly in good conditions but ends up dying). It is not clear the rates the authors have used. Furthermore, in L. 233 the authors indicate they used a 5% mortality based on previous studies, but the numbers shown in table 2 are not in agreement with these percentages and, in L. 305 the authors mentions they obtain an estimate of mortality of 4.5%. It is confusing throughout the document how mortality has been computed and the values that have been used in the different calculations.

It is important to describe the sources of mortality and clarify if the values in tables 1 and 2 correspond to at-vessel, post-release or total mortality. In my opinion, a good estimate of total mortality should include the number of tuna dead during the hauling plus the number of tuna released alive multiplied by a literature-based estimate of post-release mortality.

Also, as will be explained further below, I was not able to reproduce the expenditure estimates shown in table 2.

- Further recognition and exploration of potential biases:

It is true that a priori there is no reason for considering the surveys can be biased. On the other hand, the surveys themselves could serve to check this point, by comparing the interaction rate of interviewed anglers with the total interactions and number of participants.

As an example, the authors indicate in L295-299 that the average number of anglers onboard was 4 people, and that the number of ABFT caught per angler and season was two… Since there were, as an example, 500 anglers in 2019, on average in groups of 4 people and each group interacted on average with 2 ABFT, it would result in a total catch of 250 ABFT (assuming that each tuna is caught by a team and the four people in the team would report it as a catch). However, the total catch for that year is only of 50 fish. Even in the unlikely case two boats participate on average in the catch of a fish, it would also yield a figure well above the actual levels. It could be hypothesized that fishermen more active in the fishery, investing more time and resources are more prone to provide a response to the surveys. It that is the case, the total expenditure would be an overestimate.

- Estimates on variabiliy (e.g., sd) throughout the document would be welcome (particularly table 2). Ideally, a statistical procedure (bootstrapping, montecarlo simulations… on expenditures, post-release mortality estimates, etc.) would provide confidence intervals for the estimates.

Specific comments:

L26 “northern” should be in lower case.

L28 […] in Denmark, who often consider

L30 […] in Denmark, which peaked

L41-42 Fragmented, rephrase

L54-55 I am not very familiar with those fisheries, but I seem to remember there was a fishery for ABFT in Scandinavia with significant catches in the 1940’s. Please double-check

L61 Conservation of Atlantic Tunas

L62 Consider changing member states by Contracting Parties (it is the regular nomenclature. The EU, as an example is a party, not a MS).

L70 […] and, in 2006 and all subsequent years, […]

L73 Do not understand very well the sense of “as late as” in this context. Does it mean “as early as”?.

L78-79 The fact that Denmark and Sweden lack quotas is mentioned just two lines above.

L82 Consider removing “landing based” since it is implicit in the term commercial. To my knowledge, there are no commercial non-landing based ABFT fisheries.

L96 Some inconsistency with the lines above : 10,000 t x 69 €/kg = 0.69 billion €, but in L96 in the value is 0.9 billion €

L97 Add abbreviation (T&R program), since it is used later throughout the document

L108-110: Is this all mentioned in ref [16] or is this a conclusion of the current study?. If the latter, consider moving to the discussion section.

L118 (Table 1) Tags not reporting are, IMHO, more likely to be mortalities than the average. In any instance, the estimate of % mortality should at least be done removing them from the calculation (e.g., for Wilson et al, 2015 the calculation should be 1/(68-8)=0.7%. It should be indicated in the legend or in the main text.

Furthermore, it must be indicated (in the legend if it applies to all the studies or in an additional column if not) if the mortality is at-vessel, post-release or both.

L142 […] the fishing activities, including […]

L152 yellowfin tuna (lower case). Consider changing marlin by billfish. To my knowledge, the former does not include all the species in the family (sailfish and spearfish), although I am not sure if they are important in terms of recreational fisheries, please check.

L152-153 In other parts of the world {remove e.g.}. Only anglers adhering to minimum gear requirements (e.g. use of the circle hooks and 130 lbs test line), and with […]

L158-159 Consider rephrasing: The anglers would have had to invest […] even if no minimum gear requirements had been enforced by the tagging project, as there are

L160 […] there are no other significant recreational fisheries in Denmark which [..]

L163 […] an open recreational […]

L184 [..] contact with them

L185 […] we attempted to contact anglers…

L189 […] in the survey and, if they were,…

L195 was sent

L200 euros (lower case)

L207-2021 Consider moving to the following section on expenditures

L212-213 Merge with first sentence in the section (L202)

L226 Consider using biomass instead of weight and “ABFT mortalities due to” instead of “ABFT lost to” throughout the document.

L227 […] will give an indication of its impact on the stock.

L228-231 This part is very confusing. I first understood the authors estimated mortality rate from the literatures and applied it to the total catches, then it seemed they calculated a relative mortality rate by dividing the number of mortalities in the T&R program by the total number of tunas caught, but now they mention they calculate a relative mortality rate based on the total number of mortalities and catches in the fishery and apply it again to the total number of ABFT caught each year (does it mean there are catches and mortalities out of the T&R program). This is a central point to clarify: how mortality rate is estimated, if it includes at-vessel and post-release mortality and if the observation rate in the T&R program is not 100% and the values observed are extrapolated to the global catches.

L231-236 On top of the above, provide a rationale for the selection of 5% (it is not straightforward from the values in table 1).

L237 […] multiplied by the mean weight of ABFT caught in each respective year…

L238-239 It might be beneficial to indicate how fish measures, fish status, mortalities… were taken and recorded.

L240 SFL/CFL relationship. Use same number of decimals. If it is derived from row 2 in table 2 of Rodriguez-Marin et al (CFL=-1.887+1.0507SFL) and I am not mistaken it should yield SFL = 1.7959+0.9517CFL

L254 4,318,620€, resulting in average annual…

L256 […] not owning boats, who spent…

L264 A couple of significant issues (e.g., year 2018): (i) If the number of fish was 109 and a mortality rate of 5% was assumed, if would result in 5.45 tuna dead, not 3 (therefore, it seems they are not using a 5% mortality rate in the estimation, as stated in L 232). (ii) The total expenditures would be (227x2,356)+(75*10,734)= 1,339,862. This value, divided by the number of fish indicated in the table would be 1,339,862/3=446,620, but in the table 2 it estimates 257,671€ /tuna.

The total expenditure should also be included in the table.

Again, the value of fish dead is an actual observation (fish already dead when hauled onboard), an estimate of post-release mortality or a combination of both?.

L269 (table 3) My understanding is that these expenses are only related to the ABFT T&R activity, i.e., they would not have taken place in the absence of the T&R program. If that is the case, please note in the legend.

L274-276 I think splitting and showing the information for boat owners and non-owners was very useful. Consider doing it when mentioning incomes as well.

L304 How many fish in total, 275 (table 2) or 276?. Also, 12/275=4.4 and 12/276=4.3, not 4.5%

L305 Of the 276 ABFT, 104, 56 and 106 were caught in 2018, 2019 and 2020. Note in table 2 they are 109, 50 and 116.

L307 Indicate it is CFL

L312 At this stage, if the fishery has taken place using ICCAT research mortality allowance (BTW, it might be good to mention it in the MS) and there is not allocated quota, it should not be treated as an emerging fishery (it is not emerging, but only the result of a specific, discrete scientific action)

L315 Same as above.

L348 interested instead of invested?

L348 […] fishing in genera and, as a consequence, were more likely…

L350 […] in our analysis?

L358-359. Beginning with “multiple anglers”. As it is now, it seems all the experiential value is related to the team work and that is not possibly the case. Consider merging this statement with the following paragraph, were there is a clear link with the economic implications of the experience at the group, rather than the individual, level.

L368 Remove “entire”, since it is not comparing total incomes, but the average of the whole commercial fishery.

L370 […] trout and salmon, which are…

L383-384 Please, see note about L269 (table 3). It means the expenditures in table 3, including angling gear, boat expenses and equipment (which account for around half of the total in 2018) would not have taken place in the absence of the T&R program. Is that correct?.

L384-387. Not sure about the inference: With few exceptions, the different expenditures would have been additional inputs to the local economy regardless of the origin of the anglers. In fact, some of them (angling equipment, boat equipment…) are likely to take place elsewhere when the anglers are foreigners…

L409 Again, not sure if it is assuming a 5% mortality rate or using the observed data. Moreover, a 5% mortality would imply 3.4t fish and the 3t seems to correspond to the observed values…

A good reference for the discussion could be: Sun et al. 2019. More landings for higher profit? Inverse demand analysis of the bluefin tuna auction price in Japan and economic incentives in global bluefin tuna fisheries management. PLoS ONE 14(8): e0221147 DOI: 10.1371/journal.pone.0221147

Reviewer #2: This manuscript is an investigation of the potential economic benefits of a restored recreational bluefin tuna fishery in Denmark, especially in the Skagen peninsula. Bluefin tuna have begun reappearing off the Danish coast in the past decade and a small catch and release recreational fishery has followed. Following the collapse of bluefin in the 1960s, Denmark lost its allocation of the ICCAT Atlantic bluefin harvest, so the reemergence will remain catch and release for the foreseeable future. The authors of the manuscript surveyed Danish anglers for three years (including during the pandemic) and engaged a portion of them to engage in an unpaid recreational tag and release program, then calculated associated expenditures. The authors then project those expenditures to estimate the full value of a catch and release program to the Danish economy as well as the potential impacts on the bluefin stock from discard mortality.

The article sometimes confused economic value and economic expenditure. The economic expenditures of a recreationally landed or caught species is not directly comparable to the commercial value of a landed fish. The first is how much was spent catching the fish (a comparable commercial number would be how much was spent by a commercial fisherman doing the same) but that is not in line with the authors’ intent. In fisheries economics, we would compare consumer and producer surpluses, but that is outside the scope of this paper. There are many problems with using only expenditures–for one, it implies that the value of the fishing goes up when trip costs become more expensive (for example, if fuel prices rise), which is quite the opposite of the way an angler would perceive it. The Canadian report that the authors cite (13) is looking at charter sales (a commercial enterprise), not the recreational fishing expenditures of private anglers. They later cite Goldsmith, Scheld, and Graves (2018), who did generate consumer surplus estimates and hence economic value. The Bohnsack et al (2002) paper is more similar to what they do here, which looked at economic expenditures and economic impact. Lines 86-96, 360-372 and elsewhere that attempt to directly compare whether a fish should be allocated to the commercial or recreational sectors should be discarded or at least scaled back. Recreational fishing may well be a better use of BFT than commercial fishing, but this paper cannot show that with the data provided.

I see no issues with the biological sampling or impacts sections, with the caveat that I am an economist and not a biologist. The sampling frame looks fine and the authors did a commendable job with pushing through during the pandemic.

Less important notes:

Section on “Angler participation in the T & R program” beginning line 138–was Danish bluefin angling exclusively limited to these participants? The paper was somewhat unclear on this. Clarify. I assume that anglers can catch and release whatever they want, but I may be wrong. Lines 283-284 seem to suggest that there is a pre-existing season for BFT angling, then lines 411-412 indicate that “ a substantial recreational sector could develop in Denmark.”

Lines 171-172 –the FAO manual is on fisheries economic impacts, which is how much money is being spent in the area, make sure to use that language throughout the manuscript.

6. PLOS authors have the option to publish the peer review history of their article (what does this mean?). If published, this will include your full peer review and any attached files.

Reviewer #1: No

Reviewer #2: No

---

## [Author Response · Author response to Decision Letter 0]

2 Jun 2022

PONE-D-21-36689

Economic expenditures by recreational anglers in a recovering Atlantic bluefin tuna fishery

Response to Editor 

Dear Editor,

We would like to thank you and the reviewers for your highly constructive comments. We have found them very useful and they have helped to improve clarity and transparency of our methods, interpretations and conclusions. We have made several changes to the text and calculations to address these comments and issues. We believe that the manuscript is now sharper and easier for readers to understand and follow. Below we present detailed responses to the general and specific comments.

We hope that our manuscript may be suitable for publication in PLOS One and look forward to your decision regarding publication at earliest convenience.

Sincerely

Kristian Maar (on behalf of the co-authors)

#1

From the economics perspective, there is confusion between economic value and economic expenditure. Other technical problems concerning fisheries economics have been reported by one reviewer.

Response: 

We have considered the comments about economic value and economic expenditure/impact and have changed the manuscript accordingly. In order to clarify we have changed the phrasing or removed the mentions of economic value. Following the comments of reviewer #2, we have changed the comparisons between value and impact. The technical problems concerning fisheries economics have also been addressed – see specific replies in the individual responses below. 

#2

Another reviewer has raised several issues about the estimation of mortality rates and its use, and recommends clarification about mortality (e.g., types and sources of mortality). The paper could be improved by including a statistical framework.

Response:

We fully agree that the estimation of mortality was not presented clearly. To address this issue, we have changed the sections regarding mortality estimation and clarified how we estimate it and how it is used. See specifics below. We have also followed the recommendations to regarding estimation of uncertainties by deriving 95% upper and lower confidence limits for the mortality rates, and subsequently used these to derive the likely range of economic expenses and impact on the tuna population. Details of the method and results are provided in the manuscript, and in the individual responses below. 

#3

The manuscript needs a number of additional minor revisions and a careful revision of the English language.

Response: 

We have carefully read and addressed all the comments from the reviewers. For each minor revision and language/grammar suggestion, we have followed the reviewers’ suggestions and provide a detailed account of the resulting changes in the manuscript below. We have had the revised manuscript proof-read by a native English speaker to improve the language. 

Additional requirements

Response: 

The file names in the resubmission now follow the PLOS ONE style requirements.

Response: 

The funding information in the manuscript L470-472 has been updated to include the sentence “The funders had no role in study design, data collection and analysis, decision to publish, or preparation of the manuscript.” It now matches the ‘Financial Disclosure’. 

3. We note that Figure 1 in your submission contain map images which may be copyrighted. All PLOS content is published under the Creative Commons Attribution License (CC BY 4.0), which means that the manuscript, images, and Supporting Information files will be freely available online, and any third party is permitted to access, download, copy, distribute, and use these materials in any way, even commercially, with proper attribution. For these reasons, we cannot publish previously copyrighted maps or satellite images created using proprietary data, such as Google software (Google Maps, Street View, and Earth). For more information, see our copyright guidelines: http://journals.plos.org/plosone/s/licenses-and-copyright.

Response: 

The map figure has been removed from the submission. 

Response: 

The reference list has been reviewed and is now correct. 

Reviewers’ comments

Reviewer #1: General comments

The document presents an estimation of the potential expenditures of a recreational catch and release fishery for Atlantic bluefin tuna and the amount of bluefin tuna mortality it would imply.

The procedure is simple (an estimation of the average expense multiplied by the number of anglers and relate it to the number and weight of tuna estimated to die in the catch, tagging and release operations) and the conclusions are clear. I am not familiar with this type of studies but, in spite of its simplicity, it provides a clear an interesting message: catch and release fisheries can be a source of income more profitable than extraction fisheries by providing an alternative type (non-lethal) of ecosystem services.

In general, there are only a couple of moderately important issues that need further explanation and exploration:

Reviewer 1:

- Mortality: There are usually two types of mortality in tag and release studies: an immediate mortality (the fish is dead, or almost dead when hauled on-board) and a post-release mortality (that takes place when a fish is released seemingly in good conditions but ends up dying). It is not clear the rates the authors have used. Furthermore, in L. 233 the authors indicate they used a 5% mortality based on previous studies, but the numbers shown in table 2 are not in agreement with these percentages and, in L. 305 the authors mentions they obtain an estimate of mortality of 4.5%. It is confusing throughout the document how mortality has been computed and the values that have been used in the different calculations.

Response: 

This point has been carefully evaluated as there are multiple points of confusion about the mortality estimates in the manuscript. The reason for the confusion is in part rounding errors and a poor explanation as the reviewer mentions. The manuscript has now been corrected and the correct mortality of 4.35% is used consistently throughout. For clarification we have added a paragraph explaining the mortality observations (243-248). We have also clarified how the mortality is used for the different calculations (L251-258). We believe that these changes have improved how we estimate and use the mortality rates 

Reviewer 1:

It is important to describe the sources of mortality and clarify if the values in tables 1 and 2 correspond to at-vessel, post-release or total mortality. In my opinion, a good estimate of total mortality should include the number of tuna dead during the hauling plus the number of tuna released alive multiplied by a literature-based estimate of post-release mortality.

Response:

Agree that this needs to be more clearly described. We have made changes to the text to address this point. Please see also our reply to the previous comment. 

Reviewer 1:

Also, as will be explained further below, I was not able to reproduce the expenditure estimates shown in table 2.

Response:

The values presented in table 2 have been corrected. For further details see replies to specific comments below 

Reviewer 1:

- Further recognition and exploration of potential biases:

It is true that a priori there is no reason for considering the surveys can be biased. On the other hand, the surveys themselves could serve to check this point, by comparing the interaction rate of interviewed anglers with the total interactions and number of participants.

As an example, the authors indicate in L295-299 that the average number of anglers onboard was 4 people, and that the number of ABFT caught per angler and season was two… Since there were, as an example, 500 anglers in 2019, on average in groups of 4 people and each group interacted on average with 2 ABFT, it would result in a total catch of 250 ABFT (assuming that each tuna is caught by a team and the four people in the team would report it as a catch). However, the total catch for that year is only of 50 fish. Even in the unlikely case two boats participate on average in the catch of a fish, it would also yield a figure well above the actual levels. It could be hypothesized that fishermen more active in the fishery, investing more time and resources are more prone to provide a response to the surveys. It that is the case, the total expenditure would be an overestimate.

Response:

This is a very good point which has multiple explanations. The first is a rounding issue – the average catch angler-1 in 2019 was in fact 1.5 ABFT, with the numbers being 1.9 in 2018 and 2.1 in 2020. This has been corrected in the text (L333-334). Another reason is that compared to the relative proportion of boat-owners to non-owners in the fishery, there is an overrepresentation of boat-owners in our survey response group. As the reviewer correctly predicts – the boat-owners participate more in the fishery (they stay with the boat throughout the season) while non-owners participate less in the fishing activities. Therefore, the average boatowner will report a higher average number of tunas caught compared to the average in the total population of anglers. This is however not as big of a problem as the reviewer may think as we split the group into boat-owners and non-owners for our calculations. And therefore, higher participation and related expenditures by boat owners are not extrapolated to cover all 500 participants but only the 100 boatowners and part owners participating in 2019 while the lower expenditures of non-boat owners is used for the 400 remaining participants. This will protect against overestimation based on the responses of the most active anglers (boat owners). 

Reviewer 1:

- Estimates on variabiliy (e.g., sd) throughout the document would be welcome (particularly table 2). Ideally, a statistical procedure (bootstrapping, montecarlo simulations… on expenditures, post-release mortality estimates, etc.) would provide confidence intervals for the estimates.

Response:

Agree. We have now derived 95% upper and lower confidence limits for the mortality rates, and subsequently used these to derive the likely range of economic expenses and impact on the tuna population. Details of the method and results are provided in the manuscript.

Specific comments 

Reviewer 1:

L26 “northern” should be in lower case.

Response:

We have modified the manuscript in accordance with the suggestion of the reviewer (L26)

Reviewer 1:

L28 […] in Denmark, who often consider

Response:

We have modified the manuscript in accordance with the suggestion of the reviewer (L28)

Reviewer 1:

L30 […] in Denmark, which peaked

Response:

We have modified the manuscript in accordance with the suggestion of the reviewer (L30)

Reviewer 1:

L41-42 Fragmented, rephrase

Response:

We have modified the manuscript in accordance with the suggestion of the reviewer (L41-45)

Reviewer 1:

L54-55 I am not very familiar with those fisheries, but I seem to remember there was a fishery for ABFT in Scandinavia with significant catches in the 1940’s. Please double-check

Response: 

That is correct – changed to 1940s and 1950s (L55)

Reviewer 1:

L61 Conservation of Atlantic Tunas

Response: 

We have modified the manuscript in accordance with the suggestion of the reviewer (L61)

Reviewer 1:

L62 Consider changing member states by Contracting Parties (it is the regular nomenclature. The EU, as an example is a party, not a MS).

Response:

We have modified the manuscript in accordance with the suggestion of the reviewer (L63)

Reviewer 1:

L70 […] and, in 2006 and all subsequent years, […]

Response:

We have modified the manuscript in accordance with the suggestion of the reviewer (L70)

Reviewer 1:

L73 Do not understand very well the sense of “as late as” in this context. Does it mean “as early as”?.

Response:

Changed from ‘as late as’ to ‘up until’ 

Reviewer 1:

L78-79 The fact that Denmark and Sweden lack quotas is mentioned just two lines above.

Response:

Second mentioning removed

Reviewer 1:

L82 Consider removing “landing based” since it is implicit in the term commercial. To my knowledge, there are no commercial non-landing based ABFT fisheries.

Response:

We have modified the manuscript in accordance with the suggestion of the reviewer (L81)

Reviewer 1:

L96 Some inconsistency with the lines above : 10,000 t x 69 €/kg = 0.69 billion €, but in L96 in the value is 0.9 billion €

Response:

The report in which the numbers are found used rounded numbers which is the reason for the inconsistency. The more accurate numbers are: an average end value of ABFT in 2012 = 66.9 USD kg-1 x 13,000 metric tons = 870 million USD. This is corrected in the manuscript (L95-98)

Reviewer 1:

L97 Add abbreviation (T&R program), since it is used later throughout the document

Response:

We have modified the manuscript in accordance with the suggestion of the reviewer (L99)

Reviewer 1:

L108-110: Is this all mentioned in ref [16] or is this a conclusion of the current study?. If the latter, consider moving to the discussion section.

Response:

These findings are mentioned and summarized from the statement in reference [16] (Sutton & Ditton, 2001): “Specifically, anglers for whom fishing was more central to their lifestyle and anglers who placed lower importance on keeping fish were more likely to have a positive attitude toward catch-and-release for the Hatteras fishery and were more likely to release all bluefin tuna caught during their 1-day fishing trip.” Additionally in Table 2 in Sutton & Ditton, 2001 it is reported that the statement “I want to keep all the fish I catch” only scores 1.8 on the Likert 5 point scale. 

Reviewer 1:

L118 (Table 1) Tags not reporting are, IMHO, more likely to be mortalities than the average. In any instance, the estimate of % mortality should at least be done removing them from the calculation (e.g., for Wilson et al, 2015 the calculation should be 1/(68-8)=0.7%. It should be indicated in the legend or in the main text.

Furthermore, it must be indicated (in the legend if it applies to all the studies or in an additional column if not) if the mortality is at-vessel, post-release or both.

Response:

We thank the reviewer for pointing out this point of clarification. We have added two additional columns in table 1 indicating whether the mortalities were immediate, post-release or both. We have also recalculated the mortality percentages excluding the non-reports from the samples as suggested by the reviewer. We also added a full explanation in the table legend (L122-126) in order to clarify the issue. 

Reviewer 1:

L142 […] the fishing activities, including […]

Response:

We have modified the manuscript in accordance with the suggestion of the reviewer (L144)

Reviewer 1:

L152 yellowfin tuna (lower case). Consider changing marlin by billfish. To my knowledge, the former does not include all the species in the family (sailfish and spearfish), although I am not sure if they are important in terms of recreational fisheries, please check.

Response:

We have modified the manuscript in accordance with the suggestion of the reviewer (L154)

Reviewer 1:

L152-153 In other parts of the world {remove e.g.}. Only anglers adhering to minimum gear requirements (e.g. use of the circle hooks and 130 lbs test line), and with […]

Response:

We have modified the manuscript in accordance with the suggestion of the reviewer (L155-156)

Reviewer 1:

L158-159 Consider rephrasing: The anglers would have had to invest […] even if no minimum gear requirements had been enforced by the tagging project, as there are

Response:

We have modified the manuscript in accordance with the suggestion of the reviewer (L160-166)

Reviewer 1:

L160 […] there are no other significant recreational fisheries in Denmark which [..]

Response:

See response above.

Reviewer 1:

L163 […] an open recreational […]

Response:

We have modified the manuscript in accordance with the suggestion of the reviewer (L168)

Reviewer 1:

L184 [..] contact with them

Response:

 We have modified the manuscript in accordance with the suggestion of the reviewer (L190)

Reviewer 1:

L185 […] we attempted to contact anglers…

Response:

We have modified the manuscript in accordance with the suggestion of the reviewer (L191)

Reviewer 1:

L189 […] in the survey and, if they were,…

Response:

We have modified the manuscript in accordance with the suggestion of the reviewer (L196)

Reviewer 1:

L195 was sent

Response:

We have modified the manuscript in accordance with the suggestion of the reviewer (L201)

Reviewer 1:

L200 euros (lower case)

Response:

We have modified the manuscript in accordance with the suggestion of the reviewer (L206)

Reviewer 1:

L207-2021 Consider moving to the following section on expenditures

Response:

We have modified the manuscript in accordance with the suggestion of the reviewer (L228-232)

Reviewer 1:

L212-213 Merge with first sentence in the section (L202)

Response:

We have modified the manuscript in accordance with the suggestion of the reviewer (L208)

Reviewer 1:

L226 Consider using biomass instead of weight and “ABFT mortalities due to” instead of “ABFT lost to” throughout the document.

Response:

Weight changed to biomass in lines 236, 247, 255 and 319. ‘lost to’ changed to ‘due to’ in lines 236, 246, 254, 256 and 313. 

Reviewer 1:

L227 […] will give an indication of its impact on the stock.

Response:

We have modified the manuscript in accordance with the suggestion of the reviewer (L237)

Reviewer 1:

L228-231 This part is very confusing. I first understood the authors estimated mortality rate from the literatures and applied it to the total catches, then it seemed they calculated a relative mortality rate by dividing the number of mortalities in the T&R program by the total number of tunas caught, but now they mention they calculate a relative mortality rate based on the total number of mortalities and catches in the fishery and apply it again to the total number of ABFT caught each year (does it mean there are catches and mortalities out of the T&R program). This is a central point to clarify: how mortality rate is estimated, if it includes at-vessel and post-release mortality and if the observation rate in the T&R program is not 100% and the values observed are extrapolated to the global catches.

Response:

The confusion about the mortality estimate and related calculations is valid as we have not explained it adequately in the manuscript. We have edited the whole section about mortality (L239-275) in order to clarify this point and address all the questions raised here. 

Reviewer 1:

L231-236 On top of the above, provide a rationale for the selection of 5% (it is not straightforward from the values in table 1).

Response:

See reply to comments above. The text has been corrected for clarification. 

Reviewer 1:

L237 […] multiplied by the mean weight of ABFT caught in each respective year…

Response:

We have modified the manuscript in accordance with the suggestion of the reviewer (L247)

Reviewer 1:

L238-239 It might be beneficial to indicate how fish measures, fish status, mortalities… were taken and recorded.

Response:

We agree. Additional paragraphs explaining the details have been added (lines 248 to 250 and 253 to 258)

Reviewer 1:

L240 SFL/CFL relationship. Use same number of decimals. If it is derived from row 2 in table 2 of Rodriguez-Marin et al (CFL=-1.887+1.0507SFL) and I am not mistaken it should yield SFL = 1.7959+0.9517CFL

Response: 

Yes, the equation is from row 2 in Table 2 of Rodriguez-Marin et al. Yes, the slope coefficient for CFL should be 0.9517 instead of 0.9518 as indicated in our text. However the difference has no effect on our findings. Nevertheless, we thank the reviewer for drawing our attention to this inconsistency and have edited the text accordingly. Regarding a difference in number of decimal places in the slope and intercept terms of the equation, this is also the case in the original equation, so we have retained the same number of decimal places here.

Reviewer 1:

L254 4,318,620€, resulting in average annual…

Response:

We have modified the manuscript in accordance with the suggestion of the reviewer (L271)

Reviewer 1:

L256 […] not owning boats, who spent…

Response:

We have modified the manuscript in accordance with the suggestion of the reviewer (L273)

Reviewer 1:

L264 A couple of significant issues (e.g., year 2018): (i) If the number of fish was 109 and a mortality rate of 5% was assumed, if would result in 5.45 tuna dead, not 3 (therefore, it seems they are not using a 5% mortality rate in the estimation, as stated in L 232). (ii) The total expenditures would be (227x2,356)+(75*10,734)= 1,339,862. This value, divided by the number of fish indicated in the table would be 1,339,862/3=446,620, but in the table 2 it estimates 257,671€ /tuna.

The total expenditure should also be included in the table.

Again, the value of fish dead is an actual observation (fish already dead when hauled onboard), an estimate of post-release mortality or a combination of both?.

Response:

We thank the reviewer for spotting this error in the calculations. The actual number of tuna in 2018 was 4.52 and the resulting value is: 1,339,862/4.52 = 296,173. Table 2 has now been edited to reflect the correct values. 

The total expenditures have been added to Table 3

Clarification about the mortality has been added to the table legend (Table 2) see comments of mortality above.

Reviewer 1:

L269 (table 3) My understanding is that these expenses are only related to the ABFT T&R activity, i.e., they would not have taken place in the absence of the T&R program. If that is the case, please note in the legend.

Response:

We have modified the manuscript in accordance with the suggestion of the reviewer (L287-288)

Reviewer 1:

L274-276 I think splitting and showing the information for boat owners and non-owners was very useful. Consider doing it when mentioning incomes as well.

Response:

Good point, we have included the information for boat owners and non-owners in lines 292-295.

Reviewer 1:

L304 How many fish in total, 275 (table 2) or 276?. Also, 12/275=4.4 and 12/276=4.3, not 4.5%

Response:

We thank the reviewer for noticing this error. The numbers are 109, 50 and 116. In 2018, 2019 and 2020 respectively. Table 2 has been corrected. 

Reviewer 1:

L305 Of the 276 ABFT, 104, 56 and 106 were caught in 2018, 2019 and 2020. Note in table 2 they are 109, 50 and 116.

Response:

Once again we thank the reviewer for noticing this inconsistency – corrected according to response above. 

Reviewer 1:

L307 Indicate it is CFL

Response:

We have modified the manuscript in accordance with the suggestion of the reviewer (L326)

Reviewer 1:

L312 At this stage, if the fishery has taken place using ICCAT research mortality allowance (BTW, it might be good to mention it in the MS) and there is not allocated quota, it should not be treated as an emerging fishery (it is not emerging, but only the result of a specific, discrete scientific action)

Response:

The point is relevant and the text has now been edited so it reads “anglers participating in a scientific ABFT fishery” rather than “emerging ABFT fishery”. 

Reviewer 1:

L315 Same as above.

Response: 

The text has been edited so it is now clear that the fishery is part of the scientific T&R program, and is not an independently emerging fishery. 

Reviewer 1:

L348 interested instead of invested?

Response:

Interested is probably a better term than invested. Corrected according to the reviewers suggestion (L368)

Reviewer 1:

L348 […] fishing in genera and, as a consequence, were more likely…

Response:

We have modified the manuscript in accordance with the suggestion of the reviewer (L368-369)

Reviewer 1:

L350 […] in our analysis?

Response:

We have modified the manuscript in accordance with the suggestion of the reviewer (L370)

Reviewer 1:

L358-359. Beginning with “multiple anglers”. As it is now, it seems all the experiential value is related to the team work and that is not possibly the case. Consider merging this statement with the following paragraph, were there is a clear link with the economic implications of the experience at the group, rather than the individual, level.

Response:

We have now merged the two paragraphs, to make it clear that both the experience related to the activity likely both has an individual as well as a social character for the participating recreational fishers.

Reviewer 1:

L368 Remove “entire”, since it is not comparing total incomes, but the average of the whole commercial fishery.

Response: 

We have removed this phrase from the manuscript based on the comments from reviewer # 2. See clarification in the response to reviewer 2 below. 

Reviewer 1:

L370 […] trout and salmon, which are…

Response:

We have modified the manuscript in accordance with the suggestion of the reviewer (L370)

Reviewer 1:

L383-384 Please, see note about L269 (table 3). It means the expenditures in table 3, including angling gear, boat expenses and equipment (which account for around half of the total in 2018) would not have taken place in the absence of the T&R program. Is that correct?

Response:

Yes, that is correct. We have modified the legend in Table 3 to agree with this point and modified the text from line 160 to 170. As there are simply no other opportunities to use the highly specialized fishing equipment for other types of fishing – the expenses would not have taken place outside the T&R program, or a similar fishery for ABFT. Additionally, we specifically ask survey respondents to only report expenses ‘directly related to tuna fishing’ in each question in the survey (see survey in supplementary materials). 

Reviewer 1:

L384-387. Not sure about the inference: With few exceptions, the different expenditures would have been additional inputs to the local economy regardless of the origin of the anglers. In fact, some of them (angling equipment, boat equipment…) are likely to take place elsewhere when the anglers are foreigners…

Response:

We agree with the point raised by the reviewer here and have removed the section (L425-428) from the text. 

Reviewer 1:

L409 Again, not sure if it is assuming a 5% mortality rate or using the observed data. Moreover, a 5% mortality would imply 3.4t fish and the 3t seems to correspond to the observed values…

Response:

The value has been corrected to 4.35. For clarification on mortality see responses to comments above. 

Reviewer 1:

A good reference for the discussion could be: Sun et al. 2019. More landings for higher profit? Inverse demand analysis of the bluefin tuna auction price in Japan and economic incentives in global bluefin tuna fisheries management. PLoS ONE 14(8): e0221147 DOI: 10.1371/journal.pone.0221147

Response:

We thank the reviewer for this suggestion and have included a paragraph discussing the findings in lines 404-407. 

Reviewer #2: 

This manuscript is an investigation of the potential economic benefits of a restored recreational bluefin tuna fishery in Denmark, especially in the Skagen peninsula. Bluefin tuna have begun reappearing off the Danish coast in the past decade and a small catch and release recreational fishery has followed. Following the collapse of bluefin in the 1960s, Denmark lost its allocation of the ICCAT Atlantic bluefin harvest, so the reemergence will remain catch and release for the foreseeable future. The authors of the manuscript surveyed Danish anglers for three years (including during the pandemic) and engaged a portion of them to engage in an unpaid recreational tag and release program, then calculated associated expenditures. The authors then project those expenditures to estimate the full value of a catch and release program to the Danish economy as well as the potential impacts on the bluefin stock from discard mortality.

Reviewer #2: 

The article sometimes confused economic value and economic expenditure. The economic expenditures of a recreationally landed or caught species is not directly comparable to the commercial value of a landed fish. The first is how much was spent catching the fish (a comparable commercial number would be how much was spent by a commercial fisherman doing the same) but that is not in line with the authors’ intent. In fisheries economics, we would compare consumer and producer surpluses, but that is outside the scope of this paper. There are many problems with using only expenditures–for one, it implies that the value of the fishing goes up when trip costs become more expensive (for example, if fuel prices rise), which is quite the opposite of the way an angler would perceive it. The Canadian report that the authors cite (13) is looking at charter sales (a commercial enterprise), not the recreational fishing expenditures of private anglers. They later cite Goldsmith, Scheld, and Graves (2018), who did generate consumer surplus estimates and hence economic value. The Bohnsack et al (2002) paper is more similar to what they do here, which looked at economic expenditures and economic impact. Lines 86-96, 360-372 and elsewhere that attempt to directly compare whether a fish should be allocated to the commercial or recreational sectors should be discarded or at least scaled back. Recreational fishing may well be a better use of BFT than commercial fishing, but this paper cannot show that with the data provided.

Response:

We thank the reviewer for this important clarification, and we have modified the text for clarity according to the suggestion. Lines 86-96 have been condensed, rewritten and now avoid direct comparisons. Likewise, lines 360-372 have been reduced and only comparison with comparable data sets (i.e. economic impact data) is included. Other mentions of value has been removed or replaced with ‘impact’ see lines 356, 364 and 380.

Reviewer #2: 

I see no issues with the biological sampling or impacts sections, with the caveat that I am an economist and not a biologist. The sampling frame looks fine and the authors did a commendable job with pushing through during the pandemic.

Less important notes:

Reviewer #2: 

Section on “Angler participation in the T & R program” beginning line 138–was Danish bluefin angling exclusively limited to these participants? The paper was somewhat unclear on this. Clarify. I assume that anglers can catch and release whatever they want, but I may be wrong. Lines 283-284 seem to suggest that there is a pre-existing season for BFT angling, then lines 411-412 indicate that “ a substantial recreational sector could develop in Denmark.”

Response:

To account for this lack of clarity a new paragraph has now been added to clarify that the participants in the research project are the only recreational anglers allowed to catch ABFT in Denmark within the study period. 

Reviewer #2: 

Lines 171-172 –the FAO manual is on fisheries economic impacts, which is how much money is being spent in the area, make sure to use that language throughout the manuscript.

Response:

We have modified the manuscript using the term ‘impact’ throughout in accordance with the suggestion by the reviewer, e.g. lines 356, 364 and 380.

---

## [Decision Letter · Decision Letter 1]

8 Jul 2022

Economic expenditures by recreational anglers in a recovering Atlantic bluefin tuna fishery

PONE-D-21-36689R1

Dear Dr. Maar,

We’re pleased to inform you that your manuscript has been judged scientifically suitable for publication and will be formally accepted for publication once it meets all outstanding technical requirements.

Kind regards,

Antonio Medina Guerrero, Ph.D.

Academic Editor

PLOS ONE

Additional Editor Comments (optional):

Reviewers' comments:

Reviewer's Responses to Questions

**Comments to the Author**

1. If the authors have adequately addressed your comments raised in a previous round of review and you feel that this manuscript is now acceptable for publication, you may indicate that here to bypass the “Comments to the Author” section, enter your conflict of interest statement in the “Confidential to Editor” section, and submit your "Accept" recommendation.

Reviewer #1: All comments have been addressed

Reviewer #2: All comments have been addressed

2. Is the manuscript technically sound, and do the data support the conclusions?

Reviewer #1: Yes

Reviewer #2: Yes

3. Has the statistical analysis been performed appropriately and rigorously? 

Reviewer #1: Yes

Reviewer #2: Yes

4. Have the authors made all data underlying the findings in their manuscript fully available?

Reviewer #1: Yes

Reviewer #2: Yes

5. Is the manuscript presented in an intelligible fashion and written in standard English?

Reviewer #1: Yes

Reviewer #2: Yes

6. Review Comments to the Author

Reviewer #1: (No Response)

Reviewer #2: (No Response)

7. PLOS authors have the option to publish the peer review history of their article (what does this mean?). If published, this will include your full peer review and any attached files.

Reviewer #1: **Yes: **Francisco J. Abascal

Reviewer #2: No

---

## [Editor Report · Acceptance letter]

14 Jul 2022

PONE-D-21-36689R1 

Economic expenditures by recreational anglers in a recovering Atlantic bluefin tuna fishery 

Dear Dr. Maar:

I'm pleased to inform you that your manuscript has been deemed suitable for publication in PLOS ONE. Congratulations! Your manuscript is now with our production department. 

Kind regards, 

on behalf of

Dr. Antonio Medina Guerrero 

Academic Editor

PLOS ONE